# Accuracy of Heart Rate Measurement Under Transient States: A Validation Study of Wearables for Real-Life Monitoring

**DOI:** 10.3390/s25206319

**Published:** 2025-10-13

**Authors:** Catharina Nina Van Oost, Federica Masci, Adelien Malisse, An-Marie Schyvens, Brent Peters, Hélène Dirix, Veerle Ross, Geert Wets, An Neven, Johan Verbraecken, Jean-Marie Aerts

**Affiliations:** 1Department of Biosystems, KU Leuven, 3000 Leuven, Belgium; federica.masci@kuleuven.be (F.M.); adelien.malisse@biorics.com (A.M.); jean-marie.aerts@kuleuven.be (J.-M.A.); 2Multidisciplinary Sleep Disorders Centre, Antwerp University Hospital, 2650 Edegem, Belgium; an-marie.schyvens@uantwerpen.be (A.-M.S.); johan.verbraecken@uza.be (J.V.); 3Laboratory of Experimental Medicine and Pediatrics, University of Antwerp, 2000 Wilrijk, Belgium; 4Transportation Research Institute (IMOB), School of Transportation Sciences, Hasselt University, 3500 Hasselt, Belgium; brent.peters@uhasselt.be (B.P.); helene.dirix@uhasselt.be (H.D.); veerle.ross@uhasselt.be (V.R.); geert.wets@uhasselt.be (G.W.); an.neven@uhasselt.be (A.N.); 5Faresa, Evidence-Based Psychological Centre, 3570 Hasselt, Belgium

**Keywords:** wearable devices, validation study, heart rate, dynamic conditions, transient states, stress monitoring

## Abstract

Wearable devices are increasingly used for health and stress monitoring, yet their accuracy under dynamic, real-world conditions remains uncertain. This study validated the heart rate measurement accuracy of one chest-worn device (Zephyr BioHarness 3.0), a research-grade wrist-worn device (EmbracePlus), and five commercial wrist-worn wearable devices (Fitbit Charge 5, Fitbit Sense 2, Garmin Vivosmart 4, WHOOP 4.0, and Withings Scanwatch) against a 12-lead ECG using a 20 min protocol simulating real-life dynamics, including rest and varied-intensity walking. Device performance was evaluated across the full protocol and during transient states, defined as periods of rapid heart rate change. Accuracy and agreement were evaluated across per-second, 10 s, and 60 s resolutions. The Zephyr device showed a strong performance during all dynamic conditions. Among the wrist-worn devices, the Fitbit Charge 5 and Sense 2 showed the highest accuracy overall, while the Garmin Vivosmart 4 demonstrated greater stability during transitions. The WHOOP 4.0, Withings Scanwatch, and EmbracePlus devices performed acceptably during steady-state conditions, but were less accurate during transitions. Performance notably declined across all wrist-worn devices during transient states, with motion onset and large step changes in heart rate exacerbating measurement errors. Larger averaging windows improved accuracy by smoothing variability. The findings underscore that wrist-worn wearable devices may be better suited for average and trend heart rate monitoring rather than capturing acute dynamics. However, the Garmin and Fitbit devices showed suitable when requiring moderate accuracy during dynamic conditions. These results highlight the importance of context-specific validation and informed device selection to ensure effective use in health and stress-related applications.

## 1. Introduction

Wearable technologies are increasingly adopted in healthcare and clinical research, as they offer novel opportunities for continuous, real-world monitoring of physiological and behavioral states. While they were initially popular in sports and fitness contexts, they are increasingly accepted as tools for long-term and remote health monitoring by both clinicians and individuals and their use has expanded to chronic disease management, mental health, and preventive care [1,2]. By monitoring key biomarkers such as heart rate (HR), heart rate variability (HRV), or other electrocardiogram (ECG) metrics, as well as electrodermal activity (EDA), respiration, temperature, and sleep-related parameters, wearables can offer valuable insights into an individual’s health and well-being. Although interpreting these physiological signals in relation to complex constructs like mental stress remains methodologically challenging, prior research has demonstrated that wearable data can be used to detect relevant changes when applied appropriately [3,4,5,6].

For large-scale health monitoring, wearables must be unobtrusive, comfortable, and affordable. Consequently, commercial wrist-worn wearable devices are frequently deployed. These rely typically on 3D accelerometry and photoplethysmography (PPG), a non-invasive optical technique for measuring peripheral blood volume changes. Commercial devices generally provide access to a limited set of physiological metrics, with heart rate being the most accessible and, therefore, the most widely used in health, stress, and well-being monitoring applications [7,8,9]. While raw PPG signals and pulse rate variability (PRV) have shown value in remote health research [10,11], such data are not consistently accessible across commercial devices. Given the widespread use of heart rate in wearable-based health monitoring, ensuring accuracy is critical, particularly when the derived data are used as inputs for algorithms for stress detection, digital phenotyping, or clinical decision making.

This study is part of the MILESTONE project, which aims to monitor the health status of occupational drivers using commercial wearables. Drivers are frequently exposed to adverse working conditions, including stress, fatigue, and sedentary behavior, all of which increase their risk of developing long-term health problems [12,13]. Early identification of stress-related physiological responses may support personalized interventions to foster healthier working environments.

Researchers have extensively examined the accuracy of heart rate measurements across various wrist-worn wearable devices. A comprehensive review by Fuller et al. (2020) indicated that the most studied brands are Fitbit, Garmin, Apple, and Polar. Although no brand has been shown to consistently outperform others, Apple and Garmin are reported to be more accurate, while Fitbit tends to a slight underestimation. Additionally, the review suggests that wrist-worn wearable devices generally measure heart rate accurately in controlled settings, such as rest, and performance remains within acceptable validity thresholds in free-living conditions [14]. However, studies consistently demonstrate that accuracy declines during physical activity, particularly during higher-intensity exercise, and greater reliability observed during rest or low-intensity activities [15,16]. Hence, as measurement errors are more pronounced during fluctuations in heart rate, validation under dynamic conditions is important. Additionally, guidelines suggest that validation should be performed across different activity levels and that both steady-state and transitions, such as activity onset and recovery, should be included [17]. Nevertheless, most validation studies have focused primarily on rest, continuous exercise, or aggregations of activities, while the specific effects of transient moments, despite their relevance to real-world daily behavior, remain underexplored.

Measurement accuracy is further influenced by factors including sensor placement, contact pressure, skin tone, skin deformation, sex, BMI, and environmental variables such as ambient temperature or lighting [18]. While not all studies are consistent about their impact, these factors highlight the importance of validating wearables following best practices. In response, several consortia, including InterLive, have proposed guidelines for conducting wearable validation. These emphasize the need for transparent reporting of the considered population and the preprocessing and analysis methods used [17].

This study evaluates the accuracy of five commercial wrist-worn wearable devices, the Fitbit Charge 5, Fitbit Sense 2, Garmin Vivosmart 4, WHOOP 4.0, and the Withings Scanwatch, alongside one research-grade wrist device, the EmbracePlus, and one research-grade chest strap, the Zephyr Bioharness 3.0 (Zephyr). The commercial devices were selected for their suitability in large-scale studies, particularly in terms of their affordability and battery life. Research-grade devices served as performance benchmarks. A dynamic protocol was implemented, using a 12-lead ECG as the gold-standard reference. The study had three main objectives: (1) to validate wearables in a dynamic context representative of real-life settings, including walking at variable intensities and rest; (2) to investigate wearable performance during transient states; and (3) to assess the impact of validation resolution on wrist-wearable devices’ performances, by using the Zephyr as a high-resolution reference for second-by-second validation. In conclusion, this study aims to identify which wearables are the most suitable for dynamic, everyday settings, and examines how transient states contribute to measurement errors. Such performance differences influence practical utility of wearables and are crucial for their reliable use in applications such as stress detection and health monitoring.

## 2. Materials and Methods

### 2.1. Recruitment

A sample of 24 healthy individuals was recruited from a pool of approximately 300 students enrolled in the Master program in Human Health Engineering (Faculty of Bioscience Engineering, KU Leuven, Belgium). The sample size was based on previous validation studies of wearable heart rate monitors, which typically included 20 to 30 participants [19,20,21]. Participants were excluded if they had previously been diagnosed with respiratory or cardiac conditions that cause discomfort during moderate-intensity exercise. The study was approved by Social and Societal Ethics Committee (SMEC) of the KU Leuven (case number G-2022-5979-R2(AMD)), and formal informed consent was obtained prior to participation.

### 2.2. Study Protocol

Participants were given a spirometry mask (Metalyzer 3B, Cortex, Leipzig, Germany), a 12-lead ECG (CAM-14 module, GE Healthcare, Helsinki, Finland), a Zephyr BioHarness 3.0 (Zephyr) chest band (Medtronic, Annapolis, MD, USA), and six wrist-worn wearable devices before beginning the exercise protocol. Ten electrodes, four limb electrodes, and six chest electrodes comprise the 12-lead ECG. The participant’s limb electrodes (RA, LA, LL, and ground electrode RL) were placed on the back following an adjusted Mason–Likar placement, two on the shoulders and two above the hip bone (Figure 1). The electrodes were placed on soft tissue rather than bone tissue to form the three limb leads and three augmented limb leads. In cases where visible body hair interfered with electrode attachment, hair was gently removed, or the electrode position was minimally adjusted to ensure secure skin contact. The Zephyr was worn at the same time; hence, the optimal placement of the chest electrodes [22] was difficult, but optimal positions were approximated as closely as possible (Figure 1). These six electrodes formed the six precordial leads. The suboptimal electrode placement was less critical for this experiment, since the ECG signal was only used to calculate heart rate and not for clinical analysis. All six wrist-worn wearable devices were worn simultaneously to enable practical comparative evaluation. While every effort was made to adhere to each device’s placement guidelines, deviations from the recommended positioning were unavoidable (Figure 2). To minimize potential bias in measurement accuracy, a structured rotation scheme was implemented, systematically varying the position (proximal, middle, or distal) and the arm (left or right) across participants to ensure balanced distribution of device placement. Although wrist diameter was not controlled for, devices were worn according to manufacturer instructions, and the tightness of the straps was standardized by having the researcher fit each device to ensure consistent contact between the sensor and the skin.

Data retrieval from the 12-lead ECG, Zephyr, and EmbracePlus (EmbracePlus, Milano, Italy) was performed according to the manufacturers’ specifications. For the consumer-grade wearables, a walking activity was initiated prior to the start of the protocol and terminated afterward to facilitate data extraction and enable higher-frequency HR measurements, compared to when the devices are not in activity mode. To comply with privacy regulations and ensure data security, all devices were connected to anonymized dummy accounts and paired with a smartphone. The activity was initiated on the Garmin Vivosmart 4 (Garmin, Olathe, KS, USA), Fitbit Charge 5 (Fitbit, Boston, MA, USA), Fitbit Sense 2 (Fitbit, Boston, MA, USA), and Withings Scanwatch (Withings, Moulineaux, France). For the WHOOP 4.0 (WHOOP, Boston, MA, USA), the activity was initiated via the mobile application. All retrieved data were converted and assembled in Excel (CSV format) and stored in anonymized files. For consumer-grade devices, sampling frequency and data processing methods are proprietary and not publicly disclosed, while research-grade devices provide access to raw signals and detailed documentation of their data processing algorithms and acquisition parameters. This study focused on evaluating the processed heart rate outputs as retrieved from each device, as these reflect the data most commonly used in real-world, longitudinal applications. An overview of the data availability for each device is given in Table 1. A more detailed description of the data collection protocol and sampling frequencies, as well as an overview of the datafile formats and firmware and software versions can be found in the Appendix A. All raw data supporting the findings of this study are included in the Appendix A.

The 20 min exercise protocol, illustrated in Figure 3, comprised six sequential phases and was performed on the Pulsar 3P treadmill (h/p/cosmos sports and medical GmbH, Nussdorf, Germany). Initially, participants sit on a chair for three minutes, followed by a transition to standing still for three minutes. The chair was then removed, and participants walked on a treadmill at 4 km/h for four minutes, followed by four minutes at 6 km/h, and four minutes at 6 km/h with a 5% incline. The protocol concluded with a two-minute standing phase. The treadmill speed and incline increased gradually, except for an abrupt stop during the last phase by pushing the STOP button. Throughout the protocol, participants were requested to minimize arm movements and hold the sidebars when possible, although they were free to move their arms if necessary.

### 2.3. Data Preprocessing

Devices in the study varied in their data reporting frequency, ranging from 1 to 30 s. To standardize the analysis and assess the impact of smoothing and temporal resolution on performance, data were combined into subsets at three different frequencies: per-second, 10 s, and 60 s. Signals from the devices and the reference (ECG or Zephyr) were temporally aligned by matching the device-provided timestamps with the recorded protocol start time (accurate to the second). The aligned signals were then paired at the appropriate validation frequency. For the per-second dataset, only time points where both the reference and the device provided a measurement were retained, without applying any averaging. The 10 s dataset was created by taking each reference timestamp and averaging all device datapoints in the preceding ten seconds, thereby reflecting the ECG’s native averaging behavior. The 60 s dataset was generated by averaging both reference and device datapoints over synchronized 60 s windows. If measurements from either the device or the reference were missing, pairwise deletion was conducted, and this pair was not included in the concerning dataset. No additional signal-level preprocessing (e.g., filtering or peak detection) was applied to these outputs; thus, outliers were retained to reflect real-world settings. Detailed dataset descriptions and an example of how averaging windows were applied are available in the Appendix A.

Intentional induction of heart rate dynamics is defined as ‘transitions’ or ‘transient states’. There were five transitions, marked in yellow in Figure 4. Each transition period contained data from 10 s before until 60 s after its onset, combined forming the ‘transitions’. Transition onsets were based on predefined protocol timing and were consistent across participants and devices through timestamp alignment. Moments in which heart rate presented steady behavior are defined as ‘steady states’, encompassing all data excluding the transitions. These are marked in blue in Figure 4. Transitions were analyzed for the per-second and 10 s datasets only. They were excluded from the 60 s dataset, as this averaging interval lacks the temporal resolution needed to capture rapid changes.

### 2.4. Statistical Analyses

The performance of the devices was determined by assessing accuracy and agreement [23]. The selection of statistical metrics was based on common prevalence and criterion from relevant studies [24,25]. Two metrics were used to quantify accuracy: the mean absolute error (MAE) and mean absolute percentage error (MAPE), both calculated from the paired differences between the device and reference values in the per-second, 10 s, and 60 s datasets. To account for inter-individual variability, MAE and MAPE were computed separately for each participant. Group-level results are reported as median and interquartile range across participants. MAPE values are presented in the main manuscript, while the MAE results, which provided similar insights, are included in the Appendix A (sheet “Wilcoxon_ECG”). Three metrics were used to quantify agreement, namely the Spearman correlation coefficient (SCC), Lin’s concordance correlation coefficient (CCC), and the Bland–Altman analysis (i.e., mean bias and 95% limits of agreement (LoA)). To account for inter-subject variability, correlation coefficients were computed using a repeated-measures approach and Bland–Altman analysis was conducted via linear mixed-effects modelling with subject as a random intercept. These metrics were selected for their robustness when dealing with non-normally distributed data. For all statistical tests, a significance level of 5% was used and 95% confidence intervals (CI) were provided. Non-parametric Wilcoxon signed-rank tests were performed to evaluate whether differences between steady states and transitions were statistically significant. In the main results, the MAPE, CCC, and Bland–Altman (mean bias and LoA) are included. The results of the additional statistical metrics, which showed similar outcomes, and the specific *p*-values, omitted due to space limitations, are both provided in the Appendix A.

Various standards are used when interpreting the results. A typical boundary for the MAPE is ±10%, based on the recommendations of the American National Standards Institute Standard for cardiac monitors, heart rate meters, and alarms: “a readout error of no greater than ±10% of the input rate or ±5 bpm, whichever is greater” [26]. Typical interpretations for the strength of correlation coefficients are weak (CC ≤ 0.5), moderate (0.5–0.7), and strong (CC ≥ 0.7) [19,27]. Additionally, a stricter acceptable boundary has been previously established at 0.80 [28,29]. In line with prior research, these boundaries are adopted to interpret the validation results.

To address the study objectives, several analyses were conducted. First, overall device performance under dynamic conditions was assessed by validating consumer- and research-grade devices against the 12-lead ECG across the complete, dynamic protocol. Second, to assess device’s ability to capture the variability and dynamics of heart rate, performance during steady states and transitions was compared. This allowed us to assess performance loss during transitions relative to steady states, as well as to compare performances between devices during transitions in absolute terms. Additionally, a qualitative comparison across the five transitions was conducted to explore whether certain transitions consistently induce higher errors or whether some devices exhibit weaknesses in measuring specific transitions. Finally, to address the limitations of ECG-based validation due to its lower temporal resolution, a complementary analysis using the Zephyr was performed. Since the research-grade chest-worn device provides 1 s heart rate data, higher-resolution validation of wrist-worn wearable devices without averaging, smoothing, or pairwise deletion could be conducted.

## 3. Results

### 3.1. User Statistics and Data Collection

Twenty-four healthy university students completed the 20 min protocol. All participants were aged between 18 and 27 years. The study included 14 women, with an average age of 22.6 years (SD: 1.9) and average BMI of 22.4 (SD: 2.1), and 10 men with an average age of 23.2 years (SD: 1.7) and average BMI of 22.9 (SD: 3.7). Skin tones ranged from type one to three on the Fitzpatrick Skin Type scale. Participant demographics and details are presented in Table 2.

An overview of the realized data collection for each participant is provided in Figure 5. The 12-lead ECG recording failed for one participant due to a bad signal quality, and data from this participant were excluded from all analyses involving the ECG as reference. The Zephyr device successfully collected heart rate data from all participants. Due to some technical issues, some measurements from the wrist-worn devices failed. Detailed documentation of encountered issues can be found in the Appendix A.

For the heart rate measurements, a maximum of 28,800 s of data could have been collected per device over the full protocol. The ECG recorded a total of 3567 HR observations, the Zephyr recorded a total of 28,800 HR observations, the EmbracePlus recorded a total of 388 HR observations, the Garmin Vivosmart 4 recorded a total of 1748 HR observations, the Fitbit Charge 5 recorded a total of 11,453 HR observations, the Fitbit Sense 2 recorded a total of 12,428 HR observations, the WHOOP 4.0 recorded a total of 9762 HR observations, and the Withings Scanwatch recorded a total of 6061 HR observations. The number of paired heart rate observations available for analysis after synchronization and pairwise deletion for each dataset is reported in the results section. Note that, due to device-specific sampling frequencies, the preprocessing did not yield 1:10 or 1:60 ratios. Additionally, missingness was calculated as the proportion of expected data points, based on the sampling frequency of the reference and the analysis window, that were unavailable due to device nonrecording or missing timestamps matching those of the reference. For this calculation, only participants with a valid recording for the respective device were included, as also indicated in Figure 5. Since the EmbracePlus provided data at one-minute intervals, pairing with the ECG yielded substantially fewer data pairs than for the other devices in the 10 s dataset (531 compared to 2975 on average) and the per-second dataset (57 compared to 1921 on average). Therefore, the device was excluded from these datasets and was only evaluated at the 60 s level. Consequently, the EmbracePlus was excluded from the transition analysis.

### 3.2. Overall Performance in Measuring Heart Rate

Performances of research-grade and consumer-grade devices against the ECG reference are presented in Table 3. For each data subset, generated based on a different averaging method (i.e., 10 s, 60 s and per-second), the table shows the number of paired measurements (#paires), the calculated metrics of agreement (rmCCC and Bland–Altman), and the metric of accuracy (median MAPE with interquartile ranges). The results of the additional statistical metrics can be found in the Appendix A, sheet ‘FullProtocol_ECG’. Bland–Altman visualizations and correlation plots illustrate how the agreement behaves across heart rate ranges and whether it differs between the per-second, 10 s, and 60 s datasets for each device. These visualizations are provided in the Appendix A, sheet ‘FullProtocol_ECG_BlandAltman’.

To statistically evaluate whether the MAPE differed significantly between devices across all participants, Wilcoxon signed-rank tests were conducted for each pairwise device comparison. A Bonferroni correction was applied to account for multiple comparisons (α = 0.05/21 = 0.0024). Only comparisons with adjusted *p*-values below this threshold were considered statistically significant. Table 4 summarizes the resulting *p*-values for the 10 s dataset. Pairwise comparisons for the per-second and 60 s datasets are provided in the Appendix A, sheet ‘Wilcoxon_ECG’.

Across the full 20 min protocol, the Zephyr outperformed all wrist-worn devices, showing the highest agreement (rmCCC > 0.95) and lowest error (MAPE < 3.9%) across all datasets. Wilcoxon tests confirmed significant differences with all other devices. The Zephyr device showed a significant mean difference with the ECG (*p* < 0.001), likely due to its higher number of datapoints and narrower spread. None of the wrist-worn devices showed significant mean differences; only the Withings approached significance (*p* = 0.05–0.07), while others ranged between 0.17 and 0.90. Pairwise differences among wrist-worn wearable devices were not significant, indicating similar performance across brands. However, the metrics suggest a ranking: the Fitbit Charge 5 and Sense 2 performed best (rmCCC > 0.86; MAPE = 4.5–6.0%), followed by Garmin (lower correlation, higher bias, but MAPE < 4.4%) and EmbracePlus (higher correlation, low bias, but MAPE < 6.2%). The WHOOP 4.0 and Withings devices were the least accurate (rmCCC < 0.80; MAPE ≥ 9%). Performance rankings were consistent across temporal resolutions. Visual inspection of the Bland–Altman plots (Appendix A) did not indicate any proportional bias.

### 3.3. Performance Measuring Heart Rate Dynamics

The performances during steady states and transitions are visualized for each device by a metric of accuracy (i.e., MAPE) for the 10 s dataset in Figure 6. The boxplots are notched to indicate the approximate 95% confidence interval of the median. Results for the per-second dataset are provided in the Appendix A, sheet ‘Transitions_ECG’.

Visual inspection of Figure 6 confirmed that the Zephyr device maintained the lowest error and tightest distribution across both conditions, while wrist-worn devices showed greater spread and increased error during transitions. Furthermore, the Withings and WHOOP devices show great spread and high median MAPE values during transitions; however, the Fitbits and Garmin devices stayed under the 10% acceptability threshold.

To assess whether each device’s performance in measuring heart rate differed between steady states and transitions, Wilcoxon signed-rank tests were performed using participant-level MAPE values from the 10 s dataset. Statistically significant differences in accuracy were found for the Zephyr (*p* < 0.001), Fitbit Charge 5 (*p* = 0.0055), and Fitbit Sense 2 devices (*p* = 0.0141). No statistically significant differences were observed for the Garmin Vivosmart 4 (*p* = 0.2293), Withings Scanwatch (*p* = 0.0910), or WHOOP 4.0 devices (*p* = 0.0781).

### 3.4. Performance Depending on Type of Dynamics

For the exploratory analysis, aiming to identify device- and transition-specific measurement errors, the accuracy of each device across the five transitions was evaluated. Figure 7 presents MAPE values for each device and transition using the 10 s dataset with ECG as the reference. The boxplots are notched to indicate the approximate 95% confidence interval of the median. Results for the per-second dataset are provided in the Appendix A (‘Transitions_ECG’). For a more detailed view of agreement during each activity, Bland–Altman and correlation plots based on the 10 s dataset are shown in Figure 8.

Due to the limited data availability for certain transitions and devices, formal statistical comparisons (e.g., Wilcoxon signed-rank tests) were not performed. The results are therefore exploratory and should be interpreted with caution. Visual inspection of Figure 8 shows that Transition 2 (sitting–walking) consistently resulted in the highest MAPE values across devices, often exceeding 8–12%, followed by Transition 1 (sitting–standing), showing increased errors to a lesser extent. In contrast, Transitions 3, 4, and 5 generally showed lower MAPEs, frequently below 7%. The Zephyr device maintained low error across all transitions (MAPE < 5%), while wrist-worn devices varied: the WHOOP 4.0, and Withings Scanwatch devices displayed higher variability and frequently exceeded the 10% predefined threshold, whereas the Fitbit Charge 5, Sense 2, and Garmin Vivosmart 4 devices showed moderate increases (below 10%).

### 3.5. High-Resolution Validation

For the high-resolution validation, the wrist-worn devices were assessed against the Zephyr device using per-second data. Table 5 presents the results for each device, including the number of paired observations, agreement metrics (rmCCC and Bland–Altman analysis), and accuracy (median MAPE with interquartile range). The results of the additional statistical metrics can be found in Appendix A, sheet ‘FullProtocol_Zephyr’. Bland–Altman visualizations and correlation plots are provided in the Appendix A, sheet ‘FullProtocol_Zephyr_BlandAltman’.

None of the mean differences with the Zephyr were significant; only the Withings approached significance (*p* = 0.07), while the others ranged from 0.32 to 0.98. Consistent with the ECG-based analysis, Wilcoxon signed-rank tests were used to compare participant-level MAPEs between devices for the per-second dataset, this time using the Zephyr as reference. A Bonferroni correction was applied (α = 0.0033). Table 6 summarizes the *p*-values.

Consistent with the ECG-based results, the high-resolution validation against the Zephyr showed no statistically significant differences in MAPE between the wrist-worn devices after Bonferroni correction, suggesting comparable accuracy overall. However, ranking based on agreement and accuracy metrics showed the Fitbit Charge 5 and Sense 2 devices performing best (rmCCC = 0.85, MAPE = 7%), followed by the EmbracePlus (rmCCC = 0.81; MAPE = 7.4%) device. The Garmin Vivosmart 4 device showed reduced performance with lower agreement and higher errors (rmCCC = 0.71; MAPE = 8.0%). The WHOOP 4.0 and Withings Scanwatch devices performed worst, with rmCCC below 0.80 and MAPEs exceeding the 10% predefined threshold. Similar to the ECG-based validation, visual inspection of the Bland–Altman plots (Appendix A) did not indicate any proportional bias.

## 4. Discussion

Throughout the 20 min dynamic protocol, the Zephyr device showed the most accurate heart rate measurements, confirming the high accuracy of chest-worn wearables in real life settings and exercise [30]. In contrast, wrist-worn wearable devices exhibited greater variability, consistent with reviews reporting reduced PPG accuracy during activities or real-world conditions [24,25,31]. The statistical difference emphasizes the value of chest-worn devices in dynamic settings. Nevertheless, wrist-worn wearable devices showed relatively low, non-significant mean bias, suggesting they may capture average heart rate trends over longer durations well. Note that these non-significant biases could also originate from wider variability or higher standard errors in the wrist device measurements.

Although no statistically significant differences were found between brands, agreement and accuracy metrics over the complete protocol indicated that the Fitbit Charge 5 and Sense 2 performed better than the Garmin Vivosmart 4. To date, few validation studies have assessed the performance of the EmbracePlus, Withings Scanwatch, or WHOOP 4.0. This study is among the first to evaluate the EmbracePlus; however, its predecessor, the Empatica E4, showed notable performance drops during walking and daily activities and performed worse than the Fitbit and Garmin devices [15,32]. One study evaluated the accuracy of the inter-beat intervals (IBI) averaged over the 10 s windows of the EmbracePlus and Fitbit Sense and concluded that both showed good agreement with the ECG at rest, but found that the EmbracePlus was strongly affected by movement artefacts [33]. In this study, the EmbracePlus showed comparable performance to the Fitbits and Garmin devices when evaluated over the entire dynamic protocol, with strong correlations and trend measurements, but higher errors. The Withings Scanwatch and WHOOP 4.0 devices demonstrated the weakest performance. While WHOOP devices have been rarely validated, one study found significant declines in accuracy during resistance exercises [34]. Similarly, the Withings Scanwatch device showed reduced accuracy during walking or shopping compared to more sedentary tasks [31].

The Zephyr device demonstrated a strong performance in capturing transient heart rate behavior and exhibited robustness to movement. Although differences in accuracy between steady states and transitions were statistically significant, they remained small (median MAPE = 2.5%). Specifically, the rare drops in performance in the first transition might be attributed to greater variability in the Zephyr device’s measurements compared to the ECG. Given that the Zephyr device’s algorithm takes up to 15 s to calculate the heart rate, the possible mismatches between the Zephyr and ECG devices’ calculations are smoothed out in the 10 and 60 s datasets. Chest-worn devices are known to outperform wrist-worn wearable devices across a range of intensities [35], and the Zephyr device has previously shown a strong performance in both real-world and controlled exercise settings [30]. Additionally, other chest-worn devices have shown robust performances during activity and work-like tasks [36,37]. Our findings further confirm that the Zephyr device is robust in detecting peak and plateau behavior in the heart rate signal, apart from whether motion is present or not, since heart rate is well measured in onset, stopping, and increasing the intensity of an activity.

All wrist-worn wearable devices exhibited a decline in performance during transitions compared to steady states. Given that transitions and steady states contained activity and rest, this demonstrates that, regardless of motion artefacts, variability in the heart rate signal caused a drop in device performance. This decline aligns with known challenges in measuring heart rate during rapid intensity changes and non-steady-state conditions; in these moments, algorithmic confidence for beat detection in the PPG is lower [17]. Although transitions were not specifically analyzed, Charlton et al. (2022) found that state-of-the-art beat detection algorithms showed reduced heart rate accuracy during walking (PPG-DaLiA dataset) and psychological stress (WESAD dataset), with MAPE values below 10%. Since accuracy at baseline remained high, these declines were attributed to the presence of movement [38]. However, in real-life dynamic conditions stress and movement often cooccur, and reduced accuracy in non-steady-state conditions remains a limitation of wrist-worn wearable devices.

Among the wrist-worn wearable devices, the Garmin Vivosmart 4 showed the most stability during transitions, despite being overall less performant than the Fitbits. In contrast, the Fitbit Charge 5 and Fitbit Sense 2 devices had larger and statistically significant drops in performance during transitions. These findings are consistent with those of earlier studies, showing that Garmin devices outperform Fitbit models during treadmill activities, while Fitbits may perform better at rest [28,32]. Notably, despite the decline compared to steady states, the Garmin and Fitbit devices maintained a median MAPE below the 10% acceptability threshold during transitions, suggesting they may be suitable for applications requiring moderate accuracy during non-steady-state changes in heart rate. On the contrary, the WHOOP 4.0 and Withings Scanwatch devices showed a big drop in performance during the transitions. The lack of statistical significance may be attributable to the high variability in heart rate measurements, as reflected in the wide spread of MAPE values. The EmbracePlus was unsuitable for transient-specific analysis due to key limitations. Its 60 s reporting frequency cannot adequately capture transitions of approximately 70 s, and its conservative algorithm, which frequently rejects measurements during movement, further increased data missingness [33]. Accordingly, while the EmbracePlus device may be appropriate for estimating average heart rate under stable conditions, it is not suitable for capturing rapid heart rate changes.

A detailed analysis of specific transitions can offer valuable insights for future research and practical applications. Despite limited data and non-significant differences, patterns observed in the descriptive metrics suggest that some wrist-wearables are more susceptible to errors during rapid-onset changes in heart rate. The first transition, from sitting to standing, was less accurately captured, likely due to a sharp peak in the heart rate signal that wrist wearables failed to detect. Devices performing poorly in the first transition may be less suitable for applications requiring peak detection or capturing a precise heart rate signal in a 70 s window, for example strength training or stress monitoring [28,39]. In this study, the Fitbit and Garmin devices demonstrated acceptable performance, whereas the WHOOP and Withings devices may be less suited for such applications.

Transitions during walking showed a link between transition step size and wearable performance. Transition 2, involving the largest step increase, had the poorest performance, while Transitions 3 and 4, with more gradual changes, were measured more accurately. Additionally, Transition 2 included motion onset, whereas Transition 5, which involved a similar step size but where no motion was present, was more accurately captured. This highlights the significance of motion in transition abruptness. Wearables with lower performance in Transition 2 appear sensitive to transition abruptness, as determined by step size and motion onset. Notably, all wrist-worn devices except the Garmin Vivosmart 4 exhibited MAPE values above 10% in this transition. These wearables might be less suitable for applications with fast and/or large heart rate increases and frequent start–stop motion, such as high-intensity sports, like sprinting or interval training. These observations align with previous findings of poor peak detection and delayed response, or lag, in the Fitbit Charge 3 device during sprint activities; moreover, poorer performance has been observed in wrist-worn wearable devices during activities with complex or irregular motion patterns compared to locomotor activities, as characterized by repetitive movements [40,41,42]. Such limitations may be attributed to reduced peripheral resistance at the wrist, which dampens pulse pressure fluctuations and impairs the accuracy of PPG-based beat detection [43,44]. However, given better wearable performances in transitions 3 and 4, some wrist-worn devices may be adequate in scenarios with smaller transition step changes or sustained heart rate increases and prolonged motion, such as plateau behavior. For capturing such transitions, the Garmin and Fitbit devices appear to be well suited. Lastly, this study found that the Withings Scanwatch and WHOOP 4.0 devices performed poorly in all transitions, exhibiting either MAPE values exceeding the 10% threshold or high variability. WHOOP 4.0 only performed acceptably in Transition 5, where no motion nor peak behavior was present. These devices may therefore be unsuitable for applications requiring accurate detection of transient responses and may be better suited for general trend monitoring.

Note that transition-specific conclusions should be interpreted with caution given the limited sample size, and future research is needed to confirm these trends. Nonetheless, the insights highlight that device-specific features and limitations should be acknowledged and accounted for in both research and application to optimize the utility of these tools. Ultimately, wearable validation must be tailored to the intended use-case and application context, as emphasized by existing guidelines [24]. One initiative addressed this within stress research; the Stress in Action Wearables Database (SiA-WD) by Schoenmakers et al. (2025) offers an open-access, researcher-focused overview of over 50 commercial and research-grade wearables, rated for short- and long-term stress research [45]. Their expert scores align with our findings, considering devices that perform well in dynamic transitions, such as the Zephyr, to be better suited for short-term applications; meanwhile, those with stable average performance and good usability, such as the EmbracePlus, may be better suited for long-term monitoring.

Performance improved with larger averaging windows from per-second to 10 and 60 s datasets, due to smoothing of errors and variability. This supports previous warnings about the impact of preprocessing on device performance [24] and underscores the relevance of second-by-second data for device validation. However, such granular data remain difficult to obtain due to practical constraints like battery life and data storage for manufacturers. The improved accuracy at larger windows emphasizes that wrist wearables are generally better suited for average heart rates or long-term trends, rather than detecting transients or short-term variability. Device-specific performance differences across averaging windows, particularly for the Garmin Vivosmart 4 and Withings Scanwatch devices, further highlight the role of data reporting frequency. These devices return heart rate data at irregular and infrequent intervals, which may compromise validation accuracy. Hence, these findings underscore that both the preprocessing method and the native reporting frequency of the device can influence validation outcomes and should be carefully considered in studies.

A previous study also highlighted the limited interpretability of data due to hardware differences and emphasized the importance of providing the sampling rate for data interpretation [25]. As these device-specific parameters were not available, the ability to understand the underlying causes of performance differences was limited. There is a need for open-source tools and transparent reporting by manufacturers, especially regarding the timeframe over which instantaneous heart rate is determined. In this regard, research-grade devices are an asset as they offer access to raw physiological signals and allow insights into their processing algorithms [32,46]. However, they also come at higher costs and are not always freely available. Our findings support the observations that research-grade wearables may perform worse than or similar to consumer-grade ones but provide access to more reliable unprocessed measurements.

The findings from this validation study in a healthy, young sample are relevant for wearable-based applications in digital phenotyping, (mental) health monitoring, and particularly for the MILESTONE project, where accurate measurement of transient heart rate changes due to stress or activity is crucial. In these real-life, dynamic settings, inaccuracies in heart rate data from commercial wearables can propagate into downstream algorithms, leading to misclassifications or reduced model accuracy. To address this issue, studies should implement strategies for managing inaccurate data, including assessing signal quality and removing or cleaning unreliable segments [8]. Additionally, complementary sensor data (e.g., accelerometry) can be used to detect and remove artefactual or low-quality segments before using the signal as model input [11]. To conclude, selecting an appropriate wearable based on the specific demands of a given application, acknowledging or mitigating device limitations, is essential in ensuring the effective use of wearables.

This study had limitations that should be considered. First, the concurrent wearing of six wrist devices, spread across both arms and at varying proximities to the wrist, likely impacted device accuracy. In addition, wrist diameter and skin color were not controlled for, which may have influenced the accuracy of the wrist-worn devices. Additionally, the study’s small and homogeneous sample (healthy university students, aged 18–27, with BMI below the average for the Flemish population) limits the generalizability of the findings to broader populations, such as older adults or individuals with a higher BMI. It should also be considered that expressing error relative to heart rate implies variable absolute tolerances across heart rate ranges, and thus stricter error margins should be employed at higher intensities. Evaluating the error in R-R interval units (ms) may provide a more uniform physiological perspective: in this study, exercise intensity remained modest (walking at 4 to 6 km/h); therefore, the effect is expected to be limited. Lastly, due to the inherent variability of transitions and the limited data accessibility of wearable devices, data were not sufficient for supporting formal statistical comparisons across specific transitions. Future studies are needed to build on these exploratory findings and validate them in larger, more diverse samples.

## 5. Conclusions

This study offers valuable insights into the accuracy and limitations of various wrist- and chest-worn wearable devices in capturing heart rate under dynamic conditions. The results stress the need to account for device-specific features, validation methods, and preprocessing techniques. The superior accuracy and robustness of the Zephyr device is highlighted, particularly during transitions, reaffirming the reliability of chest-worn wearables. Among wrist-worn devices, the Fitbit Charge 5 and Sense 2 devices demonstrated the best overall accuracy, while the Garmin Vivosmart 4 device, though less accurate overall, showed the most stable performance during transitions. In contrast, the WHOOP 4.0 and Withings Scanwatch devices performed poorly, especially during transitions involving motion onset or rapid heart rate changes. The EmbracePlus, a research-grade wrist device, performed comparable to consumer-grade alternatives in stable conditions, but its low reporting frequency and high motion sensitivity limited its suitability for transition-specific analyses and applications. All wrist devices showed decreased accuracy during transitions, largely due to motion artifacts and abrupt heart rate dynamics. However, the Fitbit and Garmin devices maintained their accuracy within acceptable limits during transitions, unlike the WHOOP and Withings devices. The descriptive analyses revealed difficulties in capturing peak behavior lag in detecting abrupt increases in heart rate. Nonetheless, smaller step increases and sustained motion (i.e., plateaus) were better measured. Larger averaging windows improved performance, but second-by-second validation remains limited by practical constraints. Device-specific sampling rates and proprietary algorithms further influenced results, with limited transparency reducing interpretability. This shows the advantage of research-grade devices.

Overall, these findings demonstrate that wrist-worn wearable devices are better suited for monitoring general heart rate trends than for capturing acute, dynamic changes. However, certain devices, such as the Fitbit Charge 5, Sense 2, and Garmin Vivosmart 4, may still offer acceptable performance under dynamic conditions, whereas others, including the EmbracePlus, WHOOP 4.0, and Withings Scanwatch, are less suited for such applications. Ultimately, context-specific validation, that accounts for device capabilities, application demands, and processing methods, is necessary in ensuring the trustworthy use of wearables in stress detection and health monitoring.

## Figures and Tables

**Figure 1 sensors-25-06319-f001:**
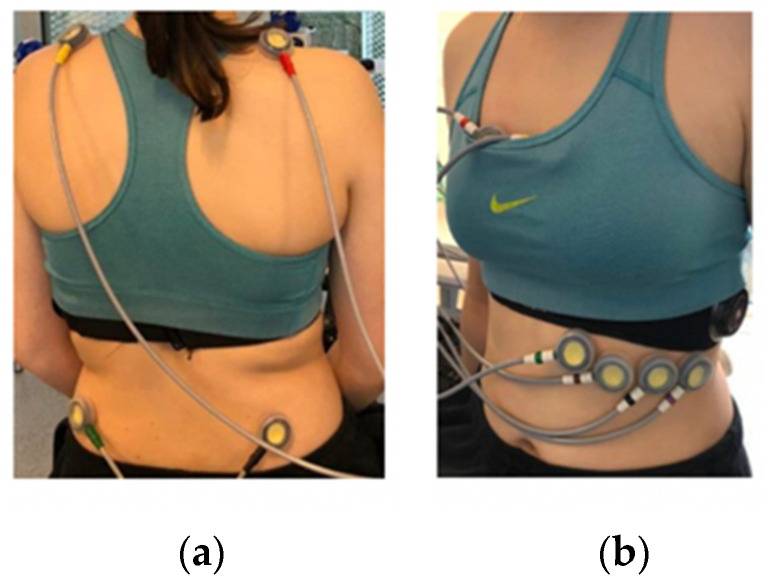
Placement of the ten electrodes from the 12-lead ECG: (**a**) placement of the four limb electrodes on the back; (**b**) placement of the six chest electrodes in combination with the Zephyr.

**Figure 2 sensors-25-06319-f002:**
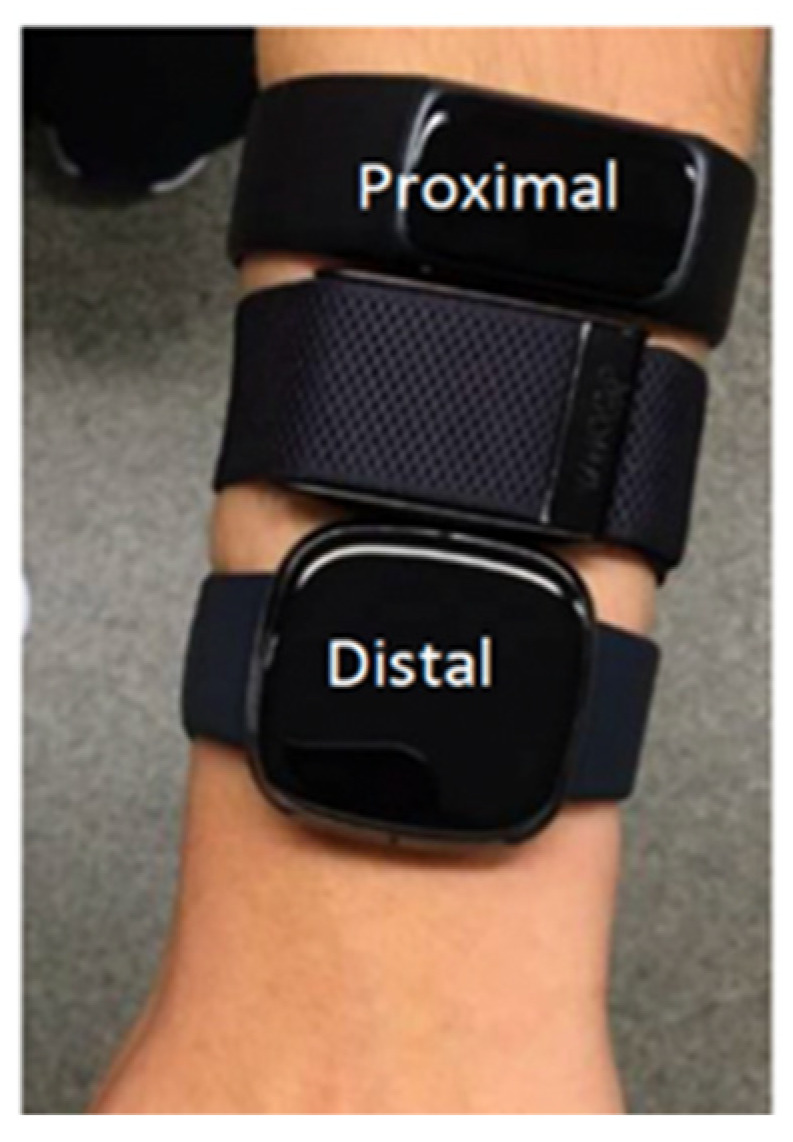
Placement of the wrist-worn wearable devices. A rotation scheme was applied. The wearables were worn as close as possible to each other. The most distal wearable was worn just above the wrist bone.

**Figure 3 sensors-25-06319-f003:**
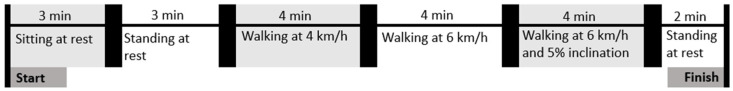
Experimental protocol of 20 min. The black lines indicate transitions or transient states.

**Figure 4 sensors-25-06319-f004:**
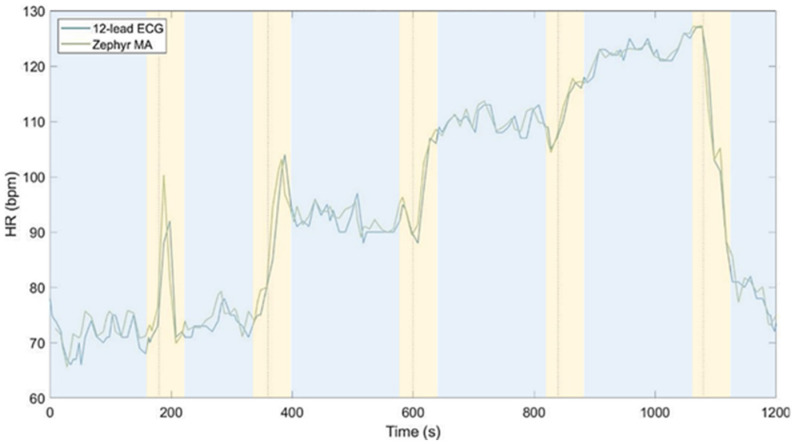
Example of 10 s moving average HR of the Zephyr (green) and 12-lead ECG (blue) for one participant. Transitions are colored in yellow and are the moments of sitting–standing, standing–walking at 4 km/h, walking at 4 km/h–walking at 6 km/h, walking at 6 km/h–walking at 6 km/h with a 5% incline, and returning to standing still. Steady states are colored in blue.

**Figure 5 sensors-25-06319-f005:**
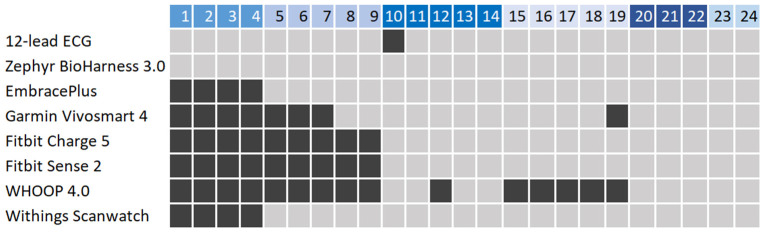
Realized data collection of all devices for each participant, numbered horizontally. Light-gray boxes indicate successfully collected data. Dark-grey boxes indicate unsuccessfully collected data. The top row shows participant numbers, color-coded by the six test days.

**Figure 6 sensors-25-06319-f006:**
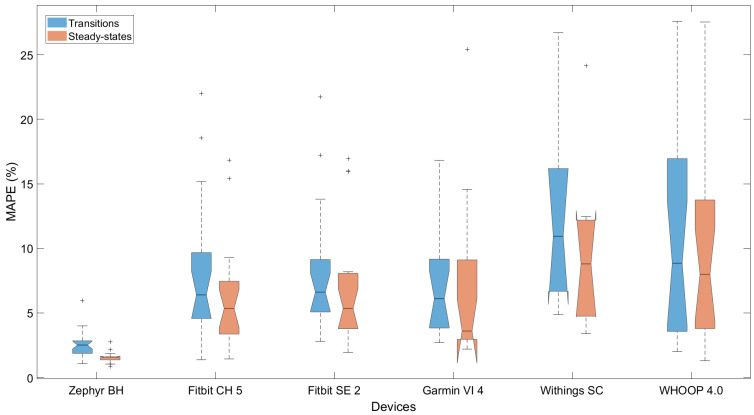
Comparison of wearable device performance during transitions (blue) and steady states (orange). Notched boxplots represent the mean absolute percentage error (MAPE) for each device using the 10 s dataset, validated against ECG.

**Figure 7 sensors-25-06319-f007:**
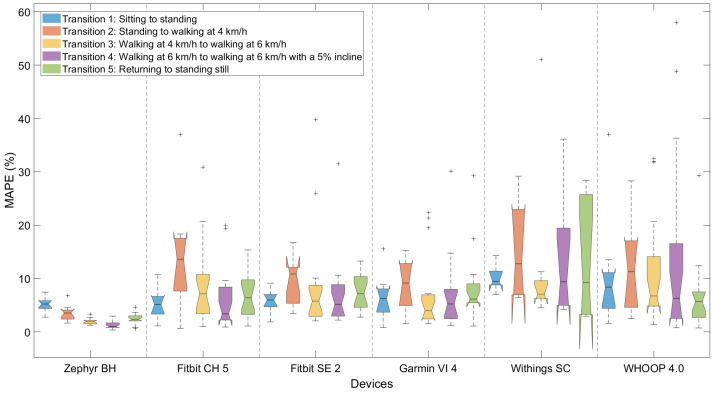
Exploratory comparison of wearable device performance for each transition. Notched boxplots represent the mean absolute percentage error (MAPE) for each device using the 10 s dataset, validated against ECG. Transition types are color-coded as indicated in the legend.

**Figure 8 sensors-25-06319-f008:**
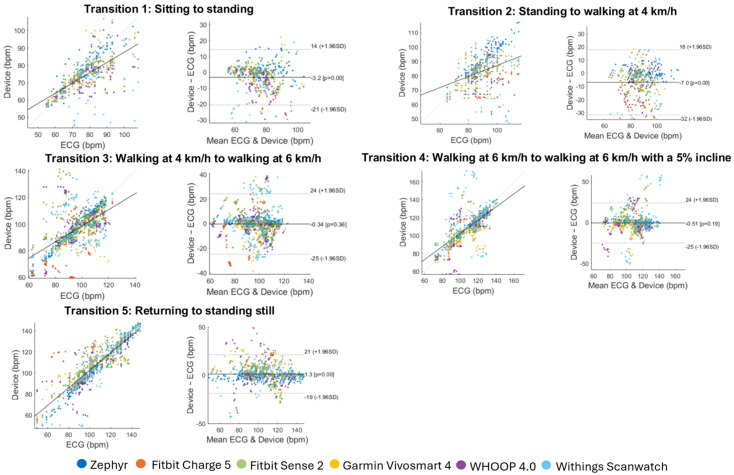
Exploratory comparison of wearable device performance for each transition. Bland–Altman visualizations and correlation plots are given per transition. The data of all devices are combined and color-coded as indicated in the legend.

**Table 1 sensors-25-06319-t001:** Overview of data availability or reporting frequency, i.e., the frequencies at which measurements were retrieved from the devices.

Device	Data Availability
12-lead ECG	Every 5 or 10 s
Zephyr Bioharness 3.0	Every second
EmbracePlus	Every minute
Garmin Vivosmart 4	Infrequent (range: 1–110 s, average: 10 s)
Fitbit Charge 5	Every 1, 2, or 3 s
Fitbit Sense 2	Every 1, 2, or 3 s
Withings Scanwatch	Infrequent (range: 1–90 s, average: 10 s)
WHOOP 4.0	Every second

**Table 2 sensors-25-06319-t002:** Participants characteristics.

Participants	Men (*n* = 11)		Women (*n* = 14)	
	Mean (SD)	Range	Mean (SD)	Range
Age (years)	23.3 (1.7)	22–27	22.6 (1.9)	18–26
Weight (kg)	78.9 (11.1)	66–100	67.9 (7.1)	56–79
Height (cm)	186.0 (7.2)	174–193	174.1 (4.8)	167–180
BMI (kg/m^2^)	22.9 (3.7)	18.1–31.2	22.4 (2.1)	18.9–25.6

**Table 3 sensors-25-06319-t003:** Validation against 12-lead ECG of research- and consumer-grade devices for the 10 s, 60 s, and per-second datasets. The metrics include repeated-measures Lin’s concordance coefficient with 95% confidence intervals (rmCCC (95%CI)), Bland–Altman analysis with mean difference and lower and higher limits of agreement (Mean Diff (lower, upper LoA)) and median MAPE with interquartile range. Statistically significant mean differences (*p* < 0.05) are indicated with *, relevant *p*-values and Bland–Altman visualizations can be found in the Appendix A.

Dataset	Device	#Pairs	Missingness (%)	rmCCC (95% CI)	Bland–Altman: Mean Diff (Lower, Upper LoA)	Median MAPE (Iqr)
10-s	Zephyr ^1^	3559	0.00%	0.98 (0.98, 0.99)	−0.30 * (−9.99, 9.32)	2.28 (0.99)
	Fitbit CH 5 ^2^	2854	9.79%	0.87 (0.85, 0.88)	0.46 (−16.52, 17.43)	5.65 (4.13)
	Fitbit SE 2 ^3^	2854	7.59%	0.86 (0.84, 0.87)	0.52 (−16.34, 17.38)	5.61 (3.53)
	Garmin VI 4 ^4^	1595	49.59%	0.77 (0.75, 0.79)	2.12 (−17.23, 21.46)	4.40 (6.18)
	Withings SC ^5^	1236	60.93%	0.68 (0.65, 0.71)	3.51 (−23.17, 30.19)	9.34 (6.59)
	WHOOP 4.0	2752	13.02%	0.79 (0.77, 0.80)	−0.39 (−24.17, 23.39)	8.52 (10.30)
60-s	Zephyr	450	0.00%	0.99 (0.98, 0.99)	−0.50 * (−5.25, 4.24)	1.45 (1.00)
	EmbracePlus	359	20.02%	0.85 (0.82, 0.88)	−0.57 (−20.42, 19.28)	6.22 (5.54)
	Fitbit CH 5	370	17.78%	0.91 (0.89, 0.92)	0.24 (−14.59, 15.06)	5.09 (4.58)
	Fitbit SE 2	378	16.00%	0.90 (0.88, 0.92)	0.33 (−14.84, 15.50)	4.55 (4.15)
	Garmin VI 4	293	34.89%	0.85 (0.81, 0.88)	1.88 (−14.48, 18.26)	3.23 (4.78)
	Withings SC	159	64.67%	0.73 (0.64, 0.79)	3.78 (−21.25, 28.82)	8.59 (7.92)
	WHOOP 4.0	390	13.33%	0.79 (0.74, 0.82)	0.96 (−24.53, 26.45)	9.99 (10.38)
Per-second	Zephyr	3567	0.00%	0.95 (0.95, 0.96)	−0.34 * (−9.99, 9.32)	3.86 (0.99)
	Fitbit CH 5	1429	59.94%	0.87 (0.85, 0.88)	0.22 (−16.59, 17.02)	6.04 (4.58)
	Fitbit SE 2	1582	55.65%	0.86 (0.84, 0.87)	−0.21 (−16.88, 16.46)	4.58 (3.53)
	Garmin VI 4	249	93.02%	0.77 (0.72, 0.82)	1.66 (−17.18, 20.49)	4.12 (6.18)
	Withings SC	1252	65.90%	0.69 (0.66, 0.72)	3.78 * (−22.88, 29.88)	8.94 (6.50)
	WHOOP 4.0	1529	57.13%	0.55 (0.51, 0.58)	0.21 (−34.03, 34.44)	11.55 (12.66)

^1^ Zephyr BioHarness 3.0; ^2^ Fitbit Charge 5; ^3^ Fitbit Sense 2; ^4^ Garmin Vivosmart 4; ^5^ Withings Scanwatch.

**Table 4 sensors-25-06319-t004:** *p*-values of Wilcoxon signed-rank tests comparing MAPEs between devices over the complete protocol (10 s dataset, validated against ECG).

Device	Fitbit CH 5	Fitbit SE 2	Garmin Vi 4	WHOOP 4.0	Withings SC
Zephyr	<0.001	<0.001	<0.001	<0.001	<0.001
Fitbit CH 5		0.715	1.000	0.046	0.199
Fitbit SE 2			0.555	0.094	0.414
Garmin Vi 4				0.087	0.298
WHOOP 4.0					0.614

**Table 5 sensors-25-06319-t005:** Validation against Zephyr of wrist-worn devices for per-second dataset.

Device	#Pairs	Missingness (%)	rmCCC (95% CI)	Bland–Altman: Mean Diff (Lower, Upper LoA)	Median MAPE (Iqr)
EmbracePlus	388	98.24%	0.81 (0.78, 0.84)	1.42 (−21.22, 24.06)	7.43 (5.25)
Fitbit CH 5	11,453	65.42%	0.85 (0.85, 0.86)	−0.65 (−19.34, 18.03)	7.17 (2.83)
Fitbit SE 2	12,428	61.03%	0.84 (0.83, 0.84)	−0.03 (−19.02, 18.95)	7.22 (4.59)
Garmin VI 4	1748	94.30%	0.71 (0.69, 0.74)	−1.75 (−23.46, 19.97)	8.04 (5.05)
Withings SC	9762	81.05%	0.64 (0.63, 0.65)	−3.54 (−30.78, 23.70)	11.20 (6.83)
WHOOP 4.0	6061	66.56%	0.78 (0.77, 0.79)	−0.42 (−25.86, 25.01)	11.37 (8.86)

**Table 6 sensors-25-06319-t006:** *p*-values of Wilcoxon signed-rank tests comparing MAPEs between devices over the complete protocol (per-second dataset, validated against Zephyr). Relevant *p*-values and Bland–Altman visualizations can be found in the Appendix A.

Device	Fitbit CH 5	Fitbit SE 2	Garmin Vi 4	WHOOP 4.0	Withings SC
EmbracePlus	0.474	0.861	0.917	0.075	0.209
Fitbit CH 5		0.603	0.377	0.024	0.066
Fitbit SE 2			0.729	0.053	0.090
Garmin Vi 4				0.232	0.331
WHOOP 4.0					0.690

## Data Availability

Data are contained within the article or the Appendix A.

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
