# Peer review of "Accuracy of Heart Rate Measurement Under Transient States: A Validation Study of Wearables for Real-Life Monitoring"

_sensors, 2025, doi:10.3390/s25206319_

Round 1

Reviewer 1 Report

Comments and Suggestions for Authors

The study addresses an important gap in wearable device validation by focusing on heart rate accuracy in "transient states" (rapid heart rate changes), which is crucial for real-life monitoring applications and under dynamic conditions. Most previous studies have focused on resting or continuous exercise. Validation was performed against a 12-lead ECG, considered the gold standard for heart rate measurement, ensuring reliable comparisons. Seven devices were compared,  evaluating accuracy and agreement at one-second, 10-second, and 60-second resolutions allows us to understand the impact of smoothing and temporal resolution on device performance.

Suggested Major Changes:

1-Variability in Data Reporting Rate: Commercial devices have variable data reporting rates and their processing methods are proprietary and not publicly disclosed. Although the study addresses this by standardizing to different resolutions (1 s, 10 s, 60 s), it remains an inherent limitation in evaluating commercial devices. In this sense, the E4 wristband provides one data point per minute. Could you explain which E4 data points are used to construct the 1-second and 10-second series? Furthermore, when there is a single E4 data point in the 60-second series, there are at least six from other wristbands, so the measurements will inevitably be noisier. In this sense, the transition analysis should not apply to the E4 band. Furthermore, I would exclude E4 from the protocol since the sampling frequency does not allow for comparissons with the rest of the devices. 

2- Additional Transition Visualization: For the exploratory transition analysis (Figure 7), consider adding Bland-Altman plots or correlation matrices for each transition if feasible, which could offer a more quantitative and detailed perspective on the accuracy and bias during each type of activity change, beyond the MAPE. Although data limitations are mentioned, an additional visual representation, even with appropriate caveats, could be useful.

3- Complete statistical information. Bland-Altman analyses cover a very wide range of heart rates. The importance of a graph of this style lies primarily in the visual inspection of the graph. In addition to providing numerical values, please consider showing Bland-Altman plots to visually determine how the agreement behaves for the different frequency ranges on each device.

Suggested Minor Changes:

1- Clarity in Zephyr Nomenclature: Some parts of the text refer to "Zephyr BioHarness 3.0" and others simply to "Zephyr." Ensure consistency in the use of the full name or a defined abbreviation at the beginning.

2- Improvement to Figure 6 Legend: The legend for Figure 6 currently reads "Performed data collection of all devices for each participant, numbered horizontally." This appears to describe a data availability figure (such as the one mentioned in Figure 5, which is not included in the provided excerpt).

3- Consistency in Writing "10-second": Sometimes it appears as "10-second" and other times as "10 second." It is recommended to standardize to "10-second" for consistency.

4- Clarification on p-values in Results: It is noted that p-values were omitted due to space limitations and can be found in Supplementary File S3. It would be helpful to at least mention the overall range of p-values or whether all were statistically significant when discussing differences, to give the reader a more complete picture without having to refer to the supplement.

5- Formatting References to Figures/Tables: Ensure that all references to figures and tables (e.g., Figure 1, Table 2) are correctly formatted and easily identifiable in the text.

Author Response

For research article: Accuracy of Heart Rate under Transient States: A Validation Study of Wearables for Real-Life Monitoring

Response to Reviewer 1 Comments

1. Summary

We would like to sincerely thank the reviewer for taking the time to carefully read our manuscript. Also, thank you for appreciating the focus on dynamic conditions and highlighting the research gap. We are grateful for the constructive and very thoughtful comments, as well as the attention to detail in identifying textual and figure-related issues We have made an effort to integrate them as good as possible and we believe that the suggestions improved the manuscript. Please find the detailed responses below.

2. Point-by-point response to Comments and Suggestions for Authors

Comments 1: Variability in Data Reporting Rate: Commercial devices have variable data reporting rates and their processing methods are proprietary and not publicly disclosed. Although the study addresses this by standardizing to different resolutions (1 s, 10 s, 60 s), it remains an inherent limitation in evaluating commercial devices. In this sense, the E4 wristband provides one data point per minute. Could you explain which E4 data points are used to construct the 1-second and 10-second series? Furthermore, when there is a single E4 data point in the 60-second series, there are at least six from other wristbands, so the measurements will inevitably be noisier. In this sense, the transition analysis should not apply to the E4 band. Furthermore, I would exclude E4 from the protocol since the sampling frequency does not allow for comparisons with the rest of the devices. 

Response 1: Thank you for the clear feedback and good suggestion. To answer the reviewer’s questions: The data points used to construct the 1-second series are the points when both the reference (ECG) and the device (E4 or EmbracePlus) provided a measurement at that timepoint. The data points used to construct the 10-second series are based on the timepoints of the reference (ECG). All the datapoints in the 10 seconds preceding that timepoint are averaged. Indeed, more clarification on how the datasets are constructed is needed. To do so, the explanation under ‘2.3. Data Pre-Processing’ in lines 182-194 was slightly adjusted and more detailed and an additional example was included in the supplementary file S1. The adjustments have been shown in red here: Devices in the study varied in their data reporting frequency, ranging from 1 to 30 seconds. To standardize the analysis and assess the impact of smoothing and temporal resolution on performance, data was combined into subsets at three different frequencies: per second, 10-second, and 60-second. Signals from the devices and the reference (ECG or Zephyr) were aligned based on start times, then paired at the appropriate validation frequency. For the per second dataset, only time points where both the reference and the device provided a measurement were retained, without applying any averaging. The 10-second dataset was created by taking each reference timestamp and averaging all device datapoints in the preceding ten seconds, thereby reflecting the ECG’s native averaging behavior. The 60-second dataset was generated by averaging both reference and device datapoints over synchronized 60-second windows. If measurements from either the device or the reference were missing, pairwise deletion was conducted and this pair was not included in the concerning dataset. No additional signal-level pre-processing (e.g., filtering or peak detection) was applied to these out-puts, thus outliers were retained to reflect real-world settings. Detailed dataset descriptions and an example of how averaging windows were applied are available in Supplementary File S1.

In supplementary file S1 was added: ‘An illustrative example of the generation of the per-second, 10-second, and 60-second datasets for the Fitbit Charge 5 in comparison with the 12-lead ECG is provided in Figure 1.’ Together with the Figure:

As the reviewer points out, the measurements of the E4 (Embraceplus) will indeed be noisier. To address this issue the reviewer suggests to remove the E4 (EmbracePlus) from the protocol. We would like to thank the reviewer for pointing this out and agree with her/his suggestion. To address this issue we removed the E4 (EmbracePlus) from the per second and 10-second analysis and in results also from the transition analysis (which is conducted with the 10-second dataset). The E4 (EmbracePlus) was removed from the transition analysis under “3.3. Performance measuring heart rate dynamics” and ” 3.4. Performance depending on type of dynamics“. The E4 (EmbracePlus) was  removed from Figure 6 and 7 and removed in the accompanying text.

However, we believe that including the E4 (Embraceplus) in the protocol does provide additional value and relevant insights. Namely, it completes the narrative that the native reporting frequency impacts the devices’ application possibilities. Additionally, in many studies, researchers also use the 1-minute heart rate measurements provided by the E4 (EmbracePlus) and not the raw PPG data. Therefore, including the E4 (EmbracePlus) to compare commercial- and research-grade wrist-worn devices in a dynamic setting is relevant. As the reviewer mentioned, there are variable data reporting rates of the commercial wearables, which can range up to 30 seconds on average. For this reason we left the E4 (EmbracePlus) part of the validation over the complete protocol under section ‘3.2. Overall performance in measuring heart rate’, but only in the 60-second resolution analysis. To address the reviewer’s suggestion we did remove the E4 (EmbracePlus) from the per second and 10-second analyses. For the removal of the E4 (EmbracePlus) from the per second and 10-second analysis the following adjustments in the text include:

-        Table 3 and 4 have been adjusted accordingly.

-        An additional note on this has been added in section “3.1. User Statistics and Data Collection” in lines 292-296: Since the EmbracePlus provided data at one-minute intervals, pairing with the ECG yielded substantially fewer data pairs than for the other devices in the 10-second dataset (531 compared to 2975 on average) and the per second dataset (57 compared to 1921 on average). Therefore, the device was excluded from these datasets and was only evaluated at the 60-second level. Consequently, the EmbracePlus was excluded from the transition analysis.

-        Removal of “except for reduced EmbracePlus performance in the per second dataset. However, this result can be neglected due to limited data” in lines 332-333.

-        Removal of “Similarly, the EmbracePlus showed reduced performance at the 10-second level, likely due to a mismatch between the analysis window and its native 60-second reporting interval.”

Adjustment in the discussion and conclusion sections includes:

-        All parts about the E4 (EmbracePlus) performance in transitions, steady states or specific transitions was removed. Specifically, within the discussion the original text: “For the EmbracePlus, the drop was statistically significant, with high errors and variability. A contributing factor is its 60-second reporting interval, which is less suited for capturing transitions (70 seconds) and results in a limited number of datapoints for analysis. Additionally, the device’s conservative algorithm tends to reject measure-ments during movement, further increasing data missingness [34]. These factors suggest that while the EmbracePlus may be appropriate for estimating average heart rate under stable conditions, it is less suitable for capturing rapid heart rate changes.” was rephrased and moved to lines 466-472 toThe EmbracePlus was unsuitable for transient-specific analysis due to key limitations. Its 60-second reporting frequency cannot adequately capture transitions of approximately 70 seconds, and its conservative algorithm, which frequently rejects measure-ments during movement, further increased data missingness [34]. Consequently, while the EmbracePlus may be appropriate for estimating average heart rate under stable conditions, it is not suitable for capturing rapid heart rate changes.”

-        A small adjustment of the conclusion about the E4’s (Embraceplus) limitations has been done in lines 588-589 by adding: The EmbracePlus, a research-grade wrist device, showed performance comparable to consumer-grade alternatives in stable conditions, but its low reporting frequency and high motion sensitivity, resulted in insufficient data for transient-specific analyses and applications.

Comments 2: Additional Transition Visualization: For the exploratory transition analysis (Figure 7), consider adding Bland-Altman plots or correlation matrices for each transition if feasible, which could offer a more quantitative and detailed perspective on the accuracy and bias during each type of activity change, beyond the MAPE. Although data limitations are mentioned, an additional visual representation, even with appropriate caveats, could be useful.

Response 2: Thank you for your suggestion, which would be a valuable addition to the manuscript. We have included the Bland Altman and correlation matrices for all the transitions. We have combined the data of all devices into the plots per transitions, but have colored the datapoints per device. An additional note has been in lines 378-379 in text under this section ‘3.4. Performance depending on type of dynamics’, namely: For a more detailed view of agreement during each activity, Bland-Altman and correlation plots based on the 10-second dataset are shown in Figure 8. Additionally, the figure below has been added under these lines:

Figure 8. Exploratory comparison of wearable device performance for each transition. Bland-Atlman visualizations and correlation plots are given per transition. Data of all devices is combined and color-coded as indicated in the legend.

Comments 3: Complete statistical information. Bland-Altman analyses cover a very wide range of heart rates. The importance of a graph of this style lies primarily in the visual inspection of the graph. In addition to providing numerical values, please consider showing Bland-Altman plots to visually determine how the agreement behaves for the different frequency ranges on each device.

Response 3: Again, thank you for your insightful and helpful suggestion. In order to include the Bland-Altman and Correlation visualisations, we have made the plots per device and color-coded the datasets (per second, 10-second and 60-second). Since the plots (2 plots per figure, 6 figures for comparison against the ECG) were rather spacious we have included them in the Supplementary File S3. We refer to the visualizations in the manuscript under lines 279-282 as: Bland–Altman visualizations and correlation plots illustrate how the agreement behaves across heart rate ranges and whether it differs between the per-second, 10-second, and 60-second datasets for each device. These visualizations are provided in Supplementary File S3, sheet ‘FullProtocol_ECG_BlandAltman’.  For consistency, we have included the Bland-Altman and correlation visualizations for validation against the Zephyr as well and referred to it in lines 360-361: Bland–Altman visualizations and correlation plots are provided in Supplementary File S3, sheet ‘FullProtocol_Zephyr_BlandAltman’.

Comments 4: Clarity in Zephyr Nomenclature: Some parts of the text refer to "Zephyr BioHarness 3.0" and others simply to "Zephyr." Ensure consistency in the use of the full name or a defined abbreviation at the beginning.

Response 4:  Yes, we have revised the text, and replaced all Zephyr Bioharnass 3.0 to Zephyr,  except in line 128-129 when describing the full name of the sensor and in Table 1 for completeness. Additionally, in table 3 ‘Zephyr BH’ has been replaced to ‘Zephyr’.

Comments 5: Improvement to Figure 6 Legend: The legend for Figure 6 currently reads "Performed data collection of all devices for each participant, numbered horizontally." This appears to describe a data availability figure (such as the one mentioned in Figure 5, which is not included in the provided excerpt).

Response 5:  Thank you for pointing this out, this mistake has been corrected and the Legend of Figure 6 is adjusted to: Comparison of wearable device performance during transitions (blue) and steady states (orange). Boxplots represent the Mean Absolute Percentage Error (MAPE) for each device using the 10-second dataset, validated against ECG.

Comments 6: Consistency in Writing "10-second": Sometimes it appears as "10-second" and other times as "10 second." It is recommended to standardize to "10-second" for consistency.

Response 6:  Again, thank you for pointing this out, throughout the text, we have adjusted 10 second to 10-second when referring to the dataset.

Comments 7: Clarification on p-values in Results: It is noted that p-values were omitted due to space limitations and can be found in Supplementary File S3. It would be helpful to at least mention the overall range of p-values or whether all were statistically significant when discussing differences, to give the reader a more complete picture without having to refer to the supplement.

Response 7: Thank you for your suggestion. If we understand the comment correctly, this refers to table 3 in section ‘3.2 Overall performance in measuring heart rate’ and table 5 in section ’3.5 High resolution validation’. To emphasize this point additional information about the p-values has been included in de text. Specifically, in lines 336-339 under section ‘3.2 Overall performance in measuring heart rate’: ‘The Zephyr showed a significant mean difference with the ECG (p < 0.001), likely due to its higher number of datapoints and narrower spread. None of the wrist-worn de-vices showed significant mean differences; only the Withings approached significance (p = 0.05-0.07), while others ranged between 0.17 and 0.90.’ Additionally, in lines 394-395, under section ’3.5 High resolution validation’: None of the mean differences with the Zephyr were significant; only the Withings approached significance (p = 0.07), while the others ranged from 0.32 to 0.98.

Comments 8: Formatting References to Figures/Tables: Ensure that all references to figures and tables (e.g., Figure 1, Table 2) are correctly formatted and easily identifiable in the text.

Response 8: Again thank you for pointing this out. We have carefully proofread the document and hope the formatting and references are correct and clear now.    

Reviewer 2 Report

Comments and Suggestions for Authors

The authors present a study evaluating the accuracy of seven wearable heart rate monitoring devices, mainly smartwatch-type, under protocols closely reflecting daily life, using a medical-grade 12-lead ECG as the reference. A notable strength of this study is its focus on transitional activity phases—a period often overlooked in prior research but critical for real-world wearable use. I found this focus to be novel and important. I hope the following suggestions may help improve the manuscript.

  1. The authors emphasize the importance of validating heart rate monitors under dynamic conditions (lines 83–85). As a relevant example, a study using a Holter ECG as a reference to validate commercially available heart rate monitors under transitions from rest to moderate exercise is available (https://doi.org/10.3390/s22239241). I suggest citing this paper in the Introduction, Results, and Discussion sections as appropriate.
  2. The accuracy of the reference heart rate is critical for performance evaluation. According to Table 1, the reference device (12-lead ECG) outputs data every 5 or 10 seconds, and the Zephyr BioHarness 3.0 outputs data every 1 second. How is the heart rate calculated in each case? For example, is it an average or most recent value within each interval? Please clarify how the representative heart rate values were derived for each reference.
  3. I recommend providing detailed information about the treadmill used in the study, including the model number and manufacturer.
  4. Please include a detailed description of how time synchronization was achieved across devices. Precise temporal alignment is essential in comparative analyses of heart rate data.
  5. In wearable ECGs, the presence of body hair at electrode attachment sites may influence signal quality. Were any measures taken to address this issue? Additionally, in the case of smartwatch-type devices, did variations in wrist diameter affect measurement accuracy?
  6. For PPG sensors, measurement accuracy can be affected by the tightness of the strap and the degree of contact between the sensor and the skin. How were these conditions standardized across participants and devices?
  7. If an absolute error margin is defined uniformly based on heart rate, the allowable deviation in milliseconds changes depending on the heart rate. For example, a 10% deviation means a tolerance of 60 ms at 100 bpm (600 ms interval) and 100 ms at 60 bpm (1000 ms interval). This results in a stricter error margin at higher heart rates. Is this approach appropriate for evaluating high-intensity activity conditions?
  8. The manuscript states that MAE and MAPE, commonly used in related studies, were employed for performance evaluation (lines 216–220). In that case, I recommend comparing your results with those of similar studies, as much as possible.

Author Response

For research article: Accuracy of Heart Rate under Transient States: A Validation Study of Wearables for Real-Life Monitoring

Response to Reviewer 2 Comments

1. Summary

We sincerely thank the reviewer for carefully and thoughtfully reading our manuscript. Thank you for appreciating the focus on transitional phases and for highlighting that this can be often overlooked in validation studies. We are particularly grateful for the valuable suggestions and for pointing us to a relevant article that has now been included. We have done our best to address all comments and integrate the feedback to improve clarity, methodological detail, and contextualization. Below, we provide detailed responses to each comment.

2. Point-by-point response to Comments and Suggestions for Authors

Comments 1: The authors emphasize the importance of validating heart rate monitors under dynamic conditions (lines 83–85). As a relevant example, a study using a Holter ECG as a reference to validate commercially available heart rate monitors under transitions from rest to moderate exercise is available (https://doi.org/10.3390/s22239241). I suggest citing this paper in the Introduction, Results, and Discussion sections as appropriate.

Response 1: We would like to thank the reviewer for the suggestion. The paper discusses the validity of a wearable heart rate monitor (smart shirt with bioelectrodes) under dynamic conditions, namely working specific movements including walking, rotating, and more. This technology is comparable to that of the Zephyr Bioharnass 3.0; therefore we included this paper as a reference. We have cited the paper in lines 434-435: “Other chest-worn devices have shown robust performances during activity and work-like tasks [37, 38]”

Comments 2: The accuracy of the reference heart rate is critical for performance evaluation. According to Table 1, the reference device (12-lead ECG) outputs data every 5 or 10 seconds, and the Zephyr BioHarness 3.0 outputs data every 1 second. How is the heart rate calculated in each case? For example, is it an average or most recent value within each interval? Please clarify how the representative heart rate values were derived for each reference.

Response 2: Thank you for pointing out this lack of clarity. When validating the Zephyr against the ECG, the dataset are constructed in the following way. Per second: select timestamps where both ECG and Zephyr provide a measurement; 10-second: select timestamps where ECG provided a measurement and average all the Zephyr measurements in the preceding 10 seconds; 60-second: average both ECG and Zephyr data over a 60 second window. We have tried to provide more clarification on how the datasets are constructed. To do so, the explanation under ‘2.3. Data Pre-Processing’ in lines 182-194 was slightly adjusted and more detailed and an additional example was included in the supplementary file S1. The adjustments have been shown in red here: Devices in the study varied in their data reporting frequency, ranging from 1 to 30 seconds. To standardize the analysis and assess the impact of smoothing and temporal resolution on performance, data was combined into subsets at three different frequencies: per second, 10-second, and 60-second. Signals from the devices and the reference (ECG or Zephyr) were aligned based on start times, then paired at the appropriate validation frequency. For the per second dataset, only time points where both the reference and the device provided a measurement were retained, without applying any averaging. The 10-second dataset was created by taking each reference timestamp and averaging all device datapoints in the preceding ten seconds, thereby reflecting the ECG’s native averaging behavior. The 60-second dataset was generated by averaging both reference and device datapoints over synchronized 60-second windows. If measurements from either the device or the reference were missing, pairwise deletion was conducted and this pair was not included in the concerning dataset. No additional signal-level pre-processing (e.g., filtering or peak detection) was applied to these out-puts, thus outliers were retained to reflect real-world settings. Detailed dataset descriptions and an example of how averaging windows were applied are available in Supplementary File S1.

In supplementary file S1 was added: ‘An illustrative example of the generation of the per-second, 10-second, and 60-second datasets for the Fitbit Charge 5 in comparison with the 12-lead ECG is provided in Figure 1.’ Together with the Figure:

Comments 3: I recommend providing detailed information about the treadmill used in the study, including the model number and manufacturer.

Response 3: Thank you for the suggestion, we have included the model and manufacturer in lines 174-175: The 20-minute exercise protocol, illustrated in Figure 3, comprised six sequential phases and was performed on the Pulsar 3P treadmill (h/p/cosmos sports & medical GmbH, Nussdorf, Germany).

Comments 4: Please include a detailed description of how time synchronization was achieved across devices. Precise temporal alignment is essential in comparative analyses of heart rate data.

Response 4: Thank you for the suggestion, this is indeed an important issue. The time synchronization was performed by aligning the start times, using the provided timestamps of the devices and the start time of the protocol (which was recorded up to seconds at the start of the protocol). We have included more clarification on this. The lines 189-190 have been adjusted from: Signals from the devices and the reference (ECG or Zephyr) were aligned based on start times, then paired at the appropriate validation frequency. To ‘Signals from the devices and the reference (ECG or Zephyr) were temporally aligned by matching the device-provided timestamps with the recorded protocol start time (accurate to the second). The aligned signals were then paired at the appropriate validation frequency.’

Comments 5: In wearable ECGs, the presence of body hair at electrode attachment sites may influence signal quality. Were any measures taken to address this issue? Additionally, in the case of smartwatch-type devices, did variations in wrist diameter affect measurement accuracy?

Response 5: Thank you for pointing this out. We only encountered body hair at the electrode attachment sites with one participant, for which we removed the hair gently and found the best position for the electrode placement (very minimal shift). We have included an additional statement in lines 135-137: In cases where visible body hair interfered with electrode attachment, hair was gently removed or the electrode position was minimally adjusted to ensure secure skin contact. The wrist diameter of the participants (in terms of the smartwatches) was not specifically measured in this study. It was only ensured that the position and tightness of the wrist-worn wearables was according to the guidelines by having the researcher put on the devices. It has been included in the statement in lines 147-150 for comments 5 and 6:

Although wrist diameter was not controlled for, devices were worn according to manufacturer instructions, and the tightness of the straps was standardized by having the researcher fit each device to ensure consistent contact between the sensor and the skin.

It has also been included in the limitations as well, in line 575-576: In addition, wrist diameter was not controlled for, which may have influenced the accuracy of the wrist-worn devices.

Comments 6: For PPG sensors, measurement accuracy can be affected by the tightness of the strap and the degree of contact between the sensor and the skin. How were these conditions standardized across participants and devices?

Response 6: Thank you for your suggestion. Indeed, this is missing from the article. The tightness of the strap was kept consistent across the participant to be according to manufacturers’ guidelines. A statement has been included in the materials and methods line 147-150, see comment 5.  

Comments 7: If an absolute error margin is defined uniformly based on heart rate, the allowable deviation in milliseconds changes depending on the heart rate. For example, a 10% deviation means a tolerance of 60 ms at 100 bpm (600 ms interval) and 100 ms at 60 bpm (1000 ms interval). This results in a stricter error margin at higher heart rates. Is this approach appropriate for evaluating high-intensity activity conditions?

Response 7: Thank you for this comment. If we understand correctly, the concern is that defining error as a percentage of heart rate leads to variable absolute tolerances across different heart rates, which may result in a stricter error margin at higher intensities. We kindly ask for clarification to ensure we have fully captured the reviewer’s point. Should we include this approach into our performance evaluations? At the same time, we would like to note that the exercise intensity in our protocol was relatively modest (walking at 4–6 km/h, with and without incline), and thus did not reach high-intensity or interval-training levels. However we did include the Bland-Altman visualizations, in the Supplementary File S3 under sheets ‘FullProtocol_ECG_BlandAltman’ and ‘FullProtocol_Zephyr_BlandAltman’, where there is no proportional bias visible. Therefore, we may say that with higher heartrates, not a more substantial deviation was visible. We would be happy to further clarify or adapt our interpretation depending on the reviewer’s suggestions.

Comments 8: The manuscript states that MAE and MAPE, commonly used in related studies, were employed for performance evaluation (lines 216–220). In that case, I recommend comparing your results with those of similar studies, as much as possible.

Response 8: Thank you for your suggestion. We have considered to include the exact MAPE and MAE values of other studies in order to quantify the differences or similarities with our result. However, unfortunately, these studies do not implement the same protocol, nor the same devices, but they use different versions (e.g. Fitbit Versa or Garmin Forerunner). Additionally, none of the studies use a similar set of brands. Therefore it did not seem relevant to include the exact MAPE and MAE values, but rather to use a descriptive comparison. However, if necessary, we are happy to make further adjustments.

Reviewer 3 Report

Comments and Suggestions for Authors

In this article authors have aimed at investigating accuracy of heart rate reporting by specific wearable devices during transient changes in heart rate. I find this an interesting article and the reported information of value.

Please see my comments below.

Were authors able to make any inferences regarding the errors associated with the device placements? Did authors investigate/detect any potential contribution to error due to wearable placements? As mentioned in the text, given the number of wrist-worn wearables, they could not all be worn as recommended by their manufacturer for optimal operation. So, I was wondering if the authors investigated whether there was any change in performance between when any given wearable was worn as recommended vs when it was not.

Were the subjects walking on a treadmill? If yes, were they holding on to something or were their hands hanging by their side? Were the subjects moving their arms during any segment of the experiment? Please provide some additional information in this regard. This is pertinent information given how movement can influence wrist-worn wearables accuracy.

In section 2.4, authors report using both MAE and MAPE metrics, but they chose to report only MAPE in the results section. Why only report MAPE?

How was sample size calculated? What is the statistical power of this study?

I recommend adding the notch feature to the boxplots in Figures 6&7 in order to also visualize the 95% CI.

Figure 6: the caption is a repeat of caption for figure 5.

Regarding Tables 3&5: Was there any proportional bias for Bland-Altman analysis? Please include this information.

Some aspects of the article need further explanation.

The number of heart rate observation for Zephyr BioHarnass 3.0 is reported to be 34091. Are these observations for total time that the device was in use, including the time before and after the 20 min duration of the experiment or were they limited to the duration of the study? If the latter, why is it a larger number than total duration of experiment in seconds, considering the 1 Hz sampling rate of the device? If the former, I recommend limiting this reporting to the 20 min of experiment for each subject.

Regarding Table 3:

Given that the 12-lead ECG device reported heart rate every 5-10 s, how is it possible that it was used as a reference for a per second comparison? Was ECG heart rate data interpolated for this purpose?

How is it possible that the values of missingness in the 10&60 second dataset segments of the table are so low for WHOOP 4 when it is missing data for 15 out of 23 possible experiments that had 12-lead ECG data?

Table 5 device column: It seems that the device names do not correspond to the rest of data in this table and the device names need to be shifted up by one and the last row seems to be a copy of the first row.

Author Response

For research article: Accuracy of Heart Rate under Transient States: A Validation Study of Wearables for Real-Life Monitoring

Response to Reviewer 3 Comments

1. Summary

We sincerely thank the reviewer for taking the time to provide such constructive and thoughtful feedback, also for expressing interest in our work. We appreciate the suggestions for adjustments to both the figures and text, which have strengthened the representation of our data and improved the overall clarity of the manuscript. We are also grateful for the reviewer’s attention to detail in identifying important mistakes and potential sources of confusion. We have carefully considered and incorporated the comments wherever possible. Detailed responses are provided below.

2. Point-by-point response to Comments and Suggestions for Authors

Comments 1: Were authors able to make any inferences regarding the errors associated with the device placements? Did authors investigate/detect any potential contribution to error due to wearable placements? As mentioned in the text, given the number of wrist-worn wearables, they could not all be worn as recommended by their manufacturer for optimal operation. So, I was wondering if the authors investigated whether there was any change in performance between when any given wearable was worn as recommended vs when it was not.

Response 1: Thank you for bringing up this important issue. We do agree that there will be a change in error depending on whether the wearable was worn on the recommended position or not. We have tried to explore the impact of the position on performance in this study, but found no significant results. Likely this is due to bigger interpersonal differences and the implementation of the rotational schema. We did not include the results in the paper. However, we do believe that there is a potential contribution to error from the improper placement, which we tried to mitigate, but we have acknowledged it as a limitation in lines 573-575: ‘First, the concurrent wearing of six wrist devices, spread across both arms and at varying proximities to the wrist, likely impacted device accuracy’.

Comments 2: Were the subjects walking on a treadmill? If yes, were they holding on to something or were their hands hanging by their side? Were the subjects moving their arms during any segment of the experiment? Please provide some additional information in this regard. This is pertinent information given how movement can influence wrist-worn wearables accuracy.

Response 2: Thank you for your suggestion. Indeed we should have included whether they were holding the side bar or whether their arms were moving. We have accordingly included additional information about this in lines 186-188, stating: Throughout the protocol, participants were requested to minimize arm movements and hold the sidebars when possible, although they were free to move their arms if necessary.

Comments 3: In section 2.4, authors report using both MAE and MAPE metrics, but they chose to report only MAPE in the results section. Why only report MAPE?.

Response 3: Thank you for noticing this error. Indeed, the MAE should also have been discussed. Since the insights from the MAE analysis were similar to the MAPE analysis, decided to not put this in the main manuscript, but add it to the Supplementary File S3 sheet ‘Wilcoxon_ECG’ where results of the statistical tests are stored. The adjustments in lines 233-235 in the text includes: MAPE values are presented in the main manuscript, while the MAE results, which provided similar insights, are included in Supplementary File S3 (sheet “Wilcoxon_ECG”).

Three new tables have been added in the Supplementary file S3, that is:

Wilcoxon Pairwise Comparisons of MAE (against ECG), Per second dataset

Embrace

Charge

Sense

Vivosmart

Whoop

Withings

Zephyr

0,041

0,024

0,001

0,210

0,000

0,001

Embrace

0,887

0,831

0,930

0,622

0,722

Charge

0,704

0,835

0,084

0,153

Sense

0,627

0,075

0,280

Vivosmart

0,129

0,199

Whoop

0,730

Wilcoxon Pairwise Comparisons of MAE (against ECG), 10-second dataset

Embrace

Charge

Sense

Vivosmart

Whoop

Withings

Zephyr

3,634E-08

7,437E-08

4,199E-08

2,760E-07

2,384E-05

4,849E-08

Embrace

7,983E-02

1,989E-01

2,823E-01

5,069E-01

7,481E-01

Charge

4,926E-01

7,549E-01

9,440E-02

1,365E-01

Sense

7,814E-01

9,440E-02

2,801E-01

Vivosmart

8,716E-02

3,315E-01

Whoop

6,904E-01

Wilcoxon Pairwise Comparisons of MAE (against ECG), 60-second dataset

Embrace

Charge

Sense

Vivosmart

Whoop

Withings

Zephyr

6,854E-06

1,222E-06

4,934E-07

3,181E-06

3,616E-05

1,470E-07

Embrace

3,353E-01

5,398E-01

2,981E-01

1,933E-01

3,972E-01

Charge

6,720E-01

8,623E-01

8,440E-02

7,493E-02

Sense

5,324E-01

7,527E-02

1,700E-01

Vivosmart

5,688E-02

7,130E-02

Whoop

6,139E-01

Comments 3: How was sample size calculated? What is the statistical power of this study?

Response 3: We did not perform a statistical test to determine the sample size before we conducted the study, but decided on the size based on previous, similar studies (as mentioned in lines 120-122 under section ‘2.1. Recruitment’. In order to answer the reviewer’s comment, we have calculated, assuming a power of 80%, alpha of 0.05 and  a device with standard deviation of 5 bpm (for example for the Fitbit Charge 5), that we can detect a difference of 4 bpm with this sample size of 24 participants. However, the statistical power depends on the variability of each sensor, therefore we have not included this in the manuscript. If the reviewer has a suggestion on how we can approach this, we are open to make any adjustments. Thank you in advance for all the support.

Comments 4: I recommend adding the notch feature to the boxplots in Figures 6&7 in order to also visualize the 95% CI.

Response 4: Thank you for this suggestion. We have applied adjusted figures 6 and 7 to include the 95% CI using the notch feature. We have added a small note on using the notched boxplots. Adjustments are in lines 376-377 under section ‘3.4. Performance depending on type of dynamics’: The boxplots are notched to indicate the approximate 95% confidence interval of the median. The image that has been adjusted is:

Figure 7. Exploratory comparison of wearable device performance for each transition. Notched boxplots represent the Mean Absolute Percentage Error (MAPE) for each device using the 10-second dataset, validated against ECG. Transition types are color-coded as indicated in the legend.

Additionally, the same adjustments have been made in section ‘3.3. Performance measuring heart rate dynamics’ with adjustment in lines 354-355: ‘The boxplots are notched to indicate the approximate 95% confidence interval of the median.’.

Figure 6. Comparison of wearable device performance during transitions (blue) and steady states (orange). Notched boxplots represent the Mean Absolute Percentage Error (MAPE) for each device using the 10-second dataset, validated against ECG.

Comments 5: Figure 6: the caption is a repeat of caption for figure 5.

Response 3: Thank you for pointing this out, I’m sorry about this mistake, the correct figure heading has been added, that is: Comparison of wearable device performance during transitions (blue) and steady states (orange). Boxplots represent the Mean Absolute Percentage Error (MAPE) for each device using the 10-second dataset, validated against ECG.

Comments 6: Regarding Tables 3&5: Was there any proportional bias for Bland-Altman analysis? Please include this information.

Response 6: Thank you for this suggestion. We have added the Bland-Altman visualizations and correlation plots for each device validated against both ECG (table 3) and Zephyr (table 5) in the Supplementary File S3 in sheet ‘FullProtocol_ECG_BlandAltman’ and ‘FullProtocol_Zephyr_BlandAltman’. Since these were 7 (against ECG) and 6 (against Zephyr) figures, we decided to not included them in the manuscript specifically, but have added an additional note to include the information, namely in lines 352-353 under section ‘3.2. Overall performance in measuring heart rate’: Visual inspection of the Bland–Altman plots (Supplementary File S3) did not indicate any proportional bias. And in lines 427-428 under section ‘3.5. High resolution validation’: Similar to the ECG-based validation, visual inspection of the Bland-Altman plots (Supplementary File S3) did not indicate any proportional bias. As an additional note, to respond to another comment from another reviewer, we did include the Bland-Altman plots of the transition data in Figure 8 under section ‘3.4. Performance depending on type of dynamics’. These plots (shown below) are included in the main manuscript and also visually, do not indicate any proportional bias.

Figure 8. Exploratory comparison of wearable device performance for each transition. Bland-Altman visualizations and correlation plots are given per transition. Data of all devices is combined and color-coded as indicated in the legend.

Comments 7: The number of heart rate observation for Zephyr BioHarnass 3.0 is reported to be 34091. Are these observations for total time that the device was in use, including the time before and after the 20 min duration of the experiment or were they limited to the duration of the study? If the latter, why is it a larger number than total duration of experiment in seconds, considering the 1 Hz sampling rate of the device? If the former, I recommend limiting this reporting to the 20 min of experiment for each subject.

Response 7: Thank you for pointing this out. Indeed, we have not limited the datapoints of the Zephyr in this overview to the duration of the study, but forgot to remove points before and after the start – and end time respectively. We have revised the data to see whether there were any missing datapoints in the duration of the study and concluded all 28800 datapoints were correctly recorded by the Zephyr. We have, accordingly, adjusted the text in line 292 to 28800 datapoints instead of 34091.

Comments 8: Regarding Table 3:

Given that the 12-lead ECG device reported heart rate every 5-10 s, how is it possible that it was used as a reference for a per second comparison? Was ECG heart rate data interpolated for this purpose?

How is it possible that the values of missingness in the 10&60 second dataset segments of the table are so low for WHOOP 4 when it is missing data for 15 out of 23 possible experiments that had 12-lead ECG data?

Response 8: Thank you for highlighting these two points. For the first point, indeed, the ECG did provide data every 5-10 seconds. We have not interpolated the data to use the ECG heart rate as reference for a per second comparison. We only selected or kept the timestamps where both the ECG (reference) and the specific device provided a measurement. There is no averaging or interpolation conducted. The difference with the 10-second dataset is that here the data of the device over the preceding 10 seconds is averaged to reflect the ECG behavior as good as possible (where the heart rate is calculated based on the IBIs of the preceding 10 seconds). We agree that more clarification on how the datasets are constructed is needed. To do so, the explanation under ‘2.3. Data Pre-Processing’ in lines 182-194 was slightly adjusted and more detailed and an additional example was included in the supplementary file S1. The adjustments have been shown in red here: Devices in the study varied in their data reporting frequency, ranging from 1 to 30 seconds. To standardize the analysis and assess the impact of smoothing and temporal resolution on performance, data was combined into subsets at three different frequencies: per second, 10-second, and 60-second. Signals from the devices and the reference (ECG or Zephyr) were aligned based on start times, then paired at the appropriate validation frequency. For the per second dataset, only time points where both the reference and the device provided a measurement were retained, without applying any averaging. The 10-second dataset was created by taking each reference timestamp and averaging all device datapoints in the preceding ten seconds, thereby reflecting the ECG’s native averaging behavior. The 60-second dataset was generated by averaging both reference and device datapoints over synchronized 60-second windows. If measurements from either the device or the reference were missing, pairwise deletion was conducted and this pair was not included in the concerning dataset. No additional signal-level pre-processing (e.g., filtering or peak detection) was applied to these out-puts, thus outliers were retained to reflect real-world settings. Detailed dataset descriptions and an example of how averaging windows were applied are available in Supplementary File S1.

In supplementary file S1 was added: ‘An illustrative example of the generation of the per-second, 10-second, and 60-second datasets for the Fitbit Charge 5 in comparison with the 12-lead ECG is provided in Figure 1.’ Together with the Figure:

We thank the reviewer for pointing out this confusion and agree that our description of missingness required better clarification. In calculating missingness, we only included participants where a recording with the device was provided. We thus did not include technical issues, or mistakes when collecting the data into account when calculating the missingness.   The intention was to capture whether the device itself experienced failures or failed to provide data at the expected frequency, rather than to reflecting participant-level data availability, which is reported in Figure 5. We have revised the text to make this distinction clearer in lines 299-302 under section ‘3.1. User Statistics and Data Collection’ has been adjusted to: ‘Additionally, missingness was calculated as the proportion of expected data points, based on the sampling frequency of the reference and the analysis window, that were unavailable due to device non-recording or missing timestamps matching those of the reference. For this calculation, only participants with a valid recording for the respective device were included, as also indicated in Figure 5.’

Comments 9: Table 5 device column: It seems that the device names do not correspond to the rest of data in this table and the device names need to be shifted up by one and the last row seems to be a copy of the first row.

Response 8: Thank you for pointing out this mistake, it was indeed shifted. I’m sorry for the error, we have made the necessary adjustment and now, Table 5 has the right amount of devices and correct information. 

Round 2

Reviewer 1 Report

Comments and Suggestions for Authors

The authors have addressed all the comments satisfactorily. I will only suggest to enlarge the text fonts in recenlty added Figure 8, to improve readibility. 

Author Response

Comment 1: The authors have addressed all the comments satisfactorily. I will only suggest to enlarge the text fonts in recenlty added Figure 8, to improve readibility. 

Response1: Thank you! We have adjusted to fonts in Figure 8 to improve readability.

Reviewer 2 Report

Comments and Suggestions for Authors

Regarding Comment 7, I thought it might be worth considering an evaluation based on absolute error relative to RRI or PPI, rather than absolute error based on heart rate. For the other comments, appropriate revisions have been made, and I have no additional remarks.

Author Response

Comment 1: Regarding Comment 7, I thought it might be worth considering an evaluation based on absolute error relative to RRI or PPI, rather than absolute error based on heart rate. For the other comments, appropriate revisions have been made, and I have no additional remarks.

Response 1: Thank you! Regarding Comment 7, we would like to thank the reviewer for the clarification. We agree that reporting error relative to RR or PP intervals (ms) provides a physiologically uniform measure across heart rate ranges, whereas percentage-based HR errors impose stricter tolerances at higher intensities. In our study, exercise intensity remained modest (walking at 4-6 km/h), so we expect this effect to be limited. Similar validation studies of commercial wearables have primarily reported percentage-based HR errors, with less emphasis on interval-based metrics. In addition, we did not have access to raw RR/PP measurements from the consumer devices, which constrained our analyses to HR-level approximations (e.g., 60000/HR). Nevertheless, we acknowledge this as an important consideration, particularly for higher-intensity protocols, and have included the following statement in the Discussion section, namely in lines 604-608: It should also be considered that expressing error relative to heart rate implies variable absolute tolerances across heart rate ranges, and thus stricter error margins at higher intensities. Evaluating the error in R-R interval units (ms) may provide a more uniform physiological perspective. In this study exercise intensity remained modest (walking at 4 to 6 km/h), therefore this effect is expected to be limited.